# How career adaptability affects entrepreneurial intention: The chain-mediated roles of entrepreneurial passion and self-efficacy

Jiping Zhang [ID]*

School of network communication, Zhejiang Yuexiu University, Shaoxing, China

* 442029227@qq.com, 20222073@zyuf.edu.cn

## Abstract

This study examined how entrepreneurial passion and entrepreneurial self-efficacy jointly mediate the relationship between career adaptability and entrepreneurial intention. A survey was conducted with 1013 Chinese university students to assess career adaptability, entrepreneurial passion, entrepreneurial self-efficacy, and entrepreneurial intention. The findings indicated that: (1) Career adaptability, entrepreneurial passion, entrepreneurial self-efficacy, and entrepreneurial intention are interconnected in ways that suggest meaningful pathways for enhancing entrepreneurial outcomes; (2) While career adaptability does not directly influence entrepreneurial intention, it has an indirect effect through entrepreneurial self-efficacy and entrepreneurial passion, highlighting the significance of these mediators; (3) Three specific mediation pathways were identified: one through entrepreneurial passion, one through entrepreneurial self-efficacy, and a third involving both factors in a chain-mediated process—where career adaptability enhances entrepreneurial passion, which then fuels entrepreneurial self-efficacy, ultimately boosting entrepreneurial intention. These findings provided deeper insight into how career adaptability shapes entrepreneurial intention and emphasized the importance of nurturing both entrepreneurial passion and entrepreneurial self-efficacy in university entrepreneurship programs. By implementing educational strategies that enhance these factors, educators can better prepare students for entrepreneurial intentions. Although focused on Chinese university students, the findings have global relevance. They offer actionable guidance for government-led entrepreneurship-education systems in higher-education institutions, particularly those in East Asia and Southeast Asia, aimed at fostering adaptive and confident future entrepreneurs.

## 1. Introduction

In a highly competitive business environment, entrepreneurship is crucial for economic development and social change [1]. It not only drives the development of advanced

**Data availability statement:** https://osf.io/dczf3/?view_only=503ff6a1e0994704b-383681f95bccabc

**Funding:** The author(s) received no specific funding for this work.

**Competing interests:** The authors have declared that no competing interests exist.

technologies, goods, and services but also creates a significant number of job opportunities [2]. Data from the Global Entrepreneurship Monitor (1999) indicated that countries with high levels of entrepreneurial activity typically experience higher economic growth rates and employment rates [3]. Against this backdrop, higher education institutions around the world are prioritizing entrepreneurship education in their strategic development. The European Union promotes the cultivation of entrepreneurial skills through the Entrepreneurship Competence Framework (EntreComp) [4,5], while U.S. universities are generally establishing entrepreneurial ecosystems [6,7]. China's "innovation and entrepreneurship" policy has also spurred a reform of the university entrepreneurship education system [8]. As an important incubator for future entrepreneurs, how universities can effectively enhance students' entrepreneurial intentions has become a common topic of research in higher education globally [9–10].

In recent years, numerous researchers have explored the formation of entrepreneurial intention from the perspective of career development [11–12]. Studies in the field of career psychology have revealed the crucial role of career adaptability in career development [13]. Savickas and Porfeli (2012) defined career adaptability as an individual's ability and psychological resources to cope with career tasks, transitions, and setbacks [14]. Savickas (2013) further pointed out in his career construction theory that career adaptability not only encompasses stable individual trait-based psychological resources but also includes the dynamic ability to adapt to changing environments, thereby effectively connecting individual proactivity with external environmental demands [15]. This theoretical perspective provided a new lens for understanding the mechanisms through which individuals form entrepreneurial intentions, particularly in entrepreneurial environments characterized by highly complex and constantly changing market contexts [16–17]. However, existing studies present discrepancies regarding the relationship between career adaptability and entrepreneurial intention: Tolentino et al. (2014) found that career adaptability positively predicts entrepreneurial intention by enhancing self-efficacy [17], whereas the study by Atitsogbe et al. (2019) indicates no significant direct relationship between the two [16]. As noted by Atitsogbe et al. (2019), if there is no direct link between career adaptability and entrepreneurial intention, then career adaptability may indirectly influence entrepreneurial intention by affecting other key factors that facilitate the realization of entrepreneurial intentions [16]. This discrepancy suggests that there may be other critical factors at play in the relationship between career adaptability and entrepreneurial intention, warranting further investigation into the mechanisms connecting the two.

To address the gap in current research, this study aims to explore the underlying mechanisms of the relationship between career adaptability and entrepreneurial intention, specifically how career adaptability influences entrepreneurial intention through mediating variables. In particular, this study focuses on the mediating roles of entrepreneurial passion and entrepreneurial self-efficacy in the relationship between career adaptability and entrepreneurial intention. The reasons for selecting these two mediating variables are threefold: First, this study is based on Social Cognitive Theory (SCT), which emphasizes the interaction between personal, environmental, and behavioral

factors [18]. Within this theoretical framework, Career adaptability in this study is viewed as a dynamic interaction between traits and environment [15,18], while entrepreneurial passion and entrepreneurial self-efficacy are considered personal factors, exploring their impact on entrepreneurial intention (behavioral factor). Second, a substantial body of literature has demonstrated that entrepreneurial passion and entrepreneurial self-efficacy play a crucial role in the formation of entrepreneurial intention [19,20]. These studies indicate that entrepreneurial passion and entrepreneurial self-efficacy not only influence the formation of entrepreneurial intention but may also play an important role in the relationship between career adaptability and entrepreneurial intention. Finally, Michl et al. (2009) point out that cognitive and emotional factors play significant roles in the entrepreneurial decision-making process [21]. Entrepreneurial passion, as an emotional factor, and entrepreneurial self-efficacy, as a cognitive factor, may both have a significant impact on entrepreneurial intention [22,23]. These factors may serve as mediators in the influence of career adaptability on entrepreneurial intention.

In summary, this study introduces a chain mediation model, grounded in social cognitive theory [24], to elucidate the mechanisms through which entrepreneurial passion (emotional dimension) and entrepreneurial self-efficacy (cognitive dimension) mediate the relationship between career adaptability and entrepreneurial intention. It investigates the mediating effects of entrepreneurial passion and entrepreneurial self-efficacy on the influence of career adaptability on entrepreneurial intention. Specifically, the study posits that when entrepreneurship is considered a viable career path, individuals with greater career adaptability are more prone to experience heightened entrepreneurial passion and increased entrepreneurial self-efficacy. These elements subsequently promote the development of entrepreneurial intention. This chain mediation model transcends existing career development models by innovatively integrating the emotional-cognitive synergy mechanism. Previous studies have often examined single mediating variables (e.g., entrepreneurial self-efficacy alone) [17], whereas we reveal the sequential pathways of career adaptability (traits-environment), entrepreneurial passion (emotion), and entrepreneurial self-efficacy (cognition). This research provides a more nuanced explanatory framework by modeling the phased mediating process of "passion (emotion) → self-efficacy (cognition)," which offers deeper theoretical insights into how "career adaptability translates into entrepreneurial intention." Furthermore, this theoretical model provides optimization insights for government-led entrepreneurial education systems in higher education institutions (such as those in East Asia and Southeast Asia), suggesting that activities to stimulate passion should be conducted before implementing efficacy enhancement training.

## 2. Literature review and research hypotheses

### 2.1 Career adaptability and entrepreneurial intention

Career adaptability refers to an individual's ability to adjust to changes in the workplace environment [25]. Savickas (2012) defines career adaptability as the capacity to manage predictable tasks within work roles and to adapt to unforeseen changes caused by variations in work content and conditions [14]. This concept specifically encompasses four dimensions: career concern, career control, career curiosity, and career confidence [14]. In recent years, some scholars have viewed career adaptability as a psychological capability for environmental adaptation, suggesting that it aids individuals in self-adjusting when facing tasks, transitions, or trauma through commitment to talent, will, and self-control, while also predicting various possibilities for their professional future [26].

Current researches suggest that career adaptability plays a significant role in influencing entrepreneurial intention. Unlike traditional career paths, entrepreneurial environments are frequently characterized by ambiguity, risk, and uncertainty [27]; thus, the capacity to adjust to market fluctuations and external conditions becomes particularly crucial [25,28]. While some studies have proposed that the effect of career adaptability on entrepreneurial intention is insignificant [16], the majority of research supports a significant correlation between career adaptability and entrepreneurial intention [25,29–31]. For instance, McKenna et al. (2016) identified a significant correlation between career adaptability and its dimensions with entrepreneurial intention [31]; a meta-analysis by Rudolph et al. (2017) revealed that career adaptability influences entrepreneurial-related outcomes [32]; and Lin (2019) determined that career adaptability has a significant positive effect on entrepreneurial intention [29].

In summary, this study suggests that career adaptability enhances individuals' capacity to manage changes and challenges within the market environment, better equipping them to handle the uncertainty and pressure associated with the entrepreneurial process, and thus increasing entrepreneurial intention. Consequently, this study proposes the following hypothesis:

H1: Career adaptability significantly and positively influences entrepreneurial intention.

## 2.2 The mediating role of entrepreneurial passion

Entrepreneurial passion generally refers to the intense interest and dedication entrepreneurs have towards their entrepreneurial endeavors, as well as the emotional experiences that emerge from this, particularly evidenced by the positive emotions and zeal exhibited by entrepreneurs throughout the entrepreneurial process [22]. Cardon et al. (2009) further suggested that entrepreneurial passion comprises two primary components: one is the positive emotional experiences linked to entrepreneurial activities, and the other is the emotional dimension associated with the entrepreneur's identity [33].

This study suggests that entrepreneurial passion might serve as a mediating factor in the connection between career adaptability and entrepreneurial intention, a notion supported by positive emotion theory and relevant entrepreneurship research theories. Initially, positive emotion theory posits that positive emotions uniquely enhance an individual's cognitive and behavioral abilities [34]. Specifically, these emotions can expand thinking patterns, boost creativity and flexibility, thereby accumulating psychological resources. This not only impacts an individual's thinking and behavior but also widens the scope of their cognition and actions [34]. In entrepreneurship research, entrepreneurial passion, as a form of positive emotion, can enrich the essential resources for entrepreneurship by broadening entrepreneurial attention and promoting cognitive flexibility [35], thus facilitating the formation of entrepreneurial intention [36]. Furthermore, Liang and Wang (2016) highlighted that the traits of entrepreneurs significantly overlap with career adaptability in dimensions such as career control, career concern, career curiosity, and career confidence [37]. Individuals with strong career adaptability are more prone to understanding and identifying with the entrepreneurial social role, thereby cultivating a sense of entrepreneurial self-identification [37]. The identity linked to the entrepreneurial role is a constituent of entrepreneurial passion [38] and a key factor in its development and maintenance [39]. Lastly, career adaptability significantly contributes to arousing entrepreneurial passion. Schellenberg and Bailis (2019) found that entrepreneurial passion is driven by task-related abilities that interact with the social environment [40]. Career adaptability is the ability to adjust to a dynamic market environment [25], and it is also a task-related ability that interacts with the social environment [26]. It can be deduced that this ability may better assist individuals in mastering core entrepreneurial activities and eliciting positive emotions, thereby driving entrepreneurial passion [40]. Therefore, career adaptability may further enhance the formation of entrepreneurial intention by influencing entrepreneurial passion. This study puts forward the hypothesis:

H2: Entrepreneurial passion mediates the effect of career adaptability on entrepreneurial intention.

## 2.3 The mediating role of entrepreneurial self-efficacy

Entrepreneurial self-efficacy pertains to an individual's subjective assessment and belief in their capacity to achieve success in entrepreneurial endeavors, essentially their self-perception of potential entrepreneurial success [41]. Studies suggest that entrepreneurial self-efficacy is pivotal in shaping entrepreneurial intentions and markedly affects an individual's inclination to pursue entrepreneurship [42–44].

Extensive research underscores the pivotal role of entrepreneurial self-efficacy as a social cognitive mechanism in shaping entrepreneurial intentions [45,46]. Moreover, career adaptability significantly affects entrepreneurial self-efficacy [47,48], indicating a potential close connection among career adaptability, entrepreneurial self-efficacy, and entrepreneurial intentions. In the realm of entrepreneurship studies, entrepreneurial self-efficacy frequently mediates the link between personal or environmental factors and entrepreneurial intentions. For example, Boyd and Vozikis (1994) discovered that

 

entrepreneurial self-efficacy can substantially moderate an individual's entrepreneurial behavior and affect the development of entrepreneurial intentions, thereby boosting the likelihood of launching new ventures [49]. Chan et al. (2012) also suggested that individuals with greater career adaptability, after undergoing multiple career transitions, accrue knowledge and relational resources that bolster their psychological self-efficacy, rendering them more prone to pursue entrepreneurship when favorable opportunities present themselves compared to their counterparts [50]. Consequently, the career adaptability of college students might sway their entrepreneurial intentions by bolstering their entrepreneurial self-efficacy [51]. A study involving Serbian university students revealed that entrepreneurial self-efficacy partially mediated the influence of career adaptability on entrepreneurial intentions [25]. Drawing on the aforementioned research, we put forth the following hypothesis:

Hypothesis 3: Entrepreneurial self-efficacy mediates the relationship between career adaptability and entrepreneurial intentions among college students.

## 2.4 The chain mediating role of entrepreneurial passion and entrepreneurial self-efficacy

Social cognitive theory emphasizes the core role of triadic reciprocal determinism (personal, behavioral, environmental) and self-efficacy [18]. In entrepreneurial contexts, the market environment often exhibits high complexity and variability. When entrepreneurship is designed as a potential career path, career adaptability helps individuals adjust to the changing market environment [25]. Career adaptability reflects an individual's proactive adjustment to the environment and can be seen as a factor in the interaction between personal traits and the environment within the SCT framework [52]. Entrepreneurial passion is the intense positive emotion and enthusiasm exhibited by entrepreneurs during the entrepreneurial process [22], which falls under individual emotional factors. As an emotional state, entrepreneurial passion represents one of the personal factors in the SCT framework [18], and it can help individuals overcome obstacles in the entrepreneurial process [28,53]. Entrepreneurial self-efficacy is an individual's subjective judgment or perception of their ability to succeed in entrepreneurship, which is a core component of SCT and belongs to cognitive factors. Bandura (1989) conceptualized the behavioral dimension of SCT as the result of the interaction among personal investment, contextual factors, and past experiences [18]. In entrepreneurship research, entrepreneurial intention is often viewed as a behavioral tendency and is considered a behavioral factor within the SCT framework [48]. Based on SCT, this study infers that individuals with high career adaptability are more likely to adjust to the changing market environment, perceive entrepreneurship as a viable career path (environmental opportunity recognition), and foster the formation of entrepreneurial intention through entrepreneurial passion and entrepreneurial self-efficacy.

Entrepreneurial passion and entrepreneurial self-efficacy are indeed closely intertwined. Cardon et al. (2013) highlighted that passion significantly contributes to boosting an individual's confidence and capability to engage in entrepreneurial endeavors [38]. Entrepreneurial passion refers to the intense positive emotion that individuals experience towards entrepreneurship [33]. Before embarking on a venture, individuals evaluate their ability to complete the anticipated tasks based on their positive or negative sentiments towards entrepreneurial activities [54]. When an individual is passionate about entrepreneurship, it can ignite their drive to participate in activities related to entrepreneurial practice, making them more inclined to seek out methods to acquire and cultivate skills pertinent to entrepreneurial activities. This, in turn, enhances their performance in the field and elevates their entrepreneurial self-efficacy [55]. Newman et al. (2019) conducted a systematic analysis of entrepreneurial passion and proposed a positive correlation between entrepreneurial passion and entrepreneurial self-efficacy [56]. Empirical studies have also substantiated this relationship. For instance, Neneh (2020) discovered through empirical research that entrepreneurial self-efficacy mediates the link between entrepreneurial passion and entrepreneurial intention among university students [57]. Boutaky and Sahib (2023) examined the influence of entrepreneurial education (an environmental factor) and entrepreneurial passion (a personal emotional factor) on entrepreneurial intention (a behavioral factor) through the lens of entrepreneurial self-efficacy (a personal cognitive factor) based on Social Cognitive Theory (SCT) [58]. Their findings indicated that entrepreneurial self-efficacy mediates the

relationship between entrepreneurial education and entrepreneurial intention, as well as between entrepreneurial passion and entrepreneurial intention [58]. Consequently, entrepreneurial passion may exert a positive influence on entrepreneurial self-efficacy.

Drawing from previous analyses, this study posits that individuals with high career adaptability are more inclined to concentrate on the entrepreneurial resources in their surroundings (career awareness), regard setbacks and obstacles encountered during entrepreneurship as chances for learning (career curiosity), and learn from their experiences, thereby adapting to the evolving market and effectively overcoming challenges. These experiences can amplify their positive engagement in entrepreneurial endeavors, reinforce their self-identity as "entrepreneurs," and foster a greater entrepreneurial passion. Such passion can ignite individuals' motivation to participate in entrepreneurial activities, which in turn bolsters their entrepreneurial self-efficacy and increases their inclination to engage in entrepreneurship. Consequently, this study proposes the following hypothesis:

H4: Entrepreneurial passion and entrepreneurial self-efficacy play a chain mediating role in the relationship between career adaptability and entrepreneurial intention.

In summary, while existing research indicates that career adaptability positively influences entrepreneurial intention [25,59], the comprehension of the relationship between these two concepts and their underlying mechanisms remains insufficient. This study, grounded in social cognitive theory, investigates the roles of two critical factors—cognition and emotion—in shaping entrepreneurial intention. Specifically, we suggest that career adaptability, as a personal factors of trait-environment interaction that adjusts to market fluctuations, impacts entrepreneurial passion and entrepreneurial self-efficacy, which are personal factors, consequently influencing entrepreneurial intention. To this end, this study develops a theoretical framework model, employing entrepreneurial passion and entrepreneurial self-efficacy as mediating variables, to systematically elucidate the mechanism through which career adaptability influences entrepreneurial intention (see Fig 1).

## 3. Methods

### 3.1 Ethics approval

The study received written ethical approval of the Academic Ethics Committee of Hainan Technology and Business College (HGS-2021–8). The Declaration of Helsinki and ethical standards were followed [60]. The recruitment period for this study ran from April 1, 2023, to June 1, 2023. Participants expressed informed consent and agreed to participate in this study, providing their written informed consent before data collection began.

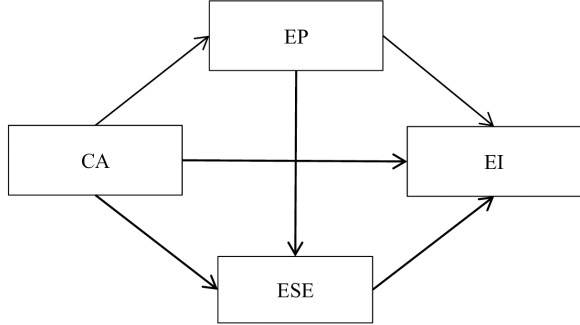

**Fig 1. The hypothetical model of EP and ESE between CA and EI.**

## 3.2 Participants and procedures

This study primarily collected data from college students in Zhejiang Province, China, mainly because Zhejiang is a region known for its advanced entrepreneurship education, with a relatively developed economy and a strong atmosphere for innovation and entrepreneurship. We selected three schools designated by the Chinese Ministry of Education as demonstration schools for career and entrepreneurship education. All these schools have offered a course in 'Career Life Planning,' and all students have completed this course, demonstrating good career adaptability (CA).This study concentrates on a sample of higher education institutions in Zhejiang Province for several reasons. Firstly, the entrepreneurship education curriculum is notably representative. Zhejiang Province is one of China's pioneering national-level experimental zones for innovation and entrepreneurship education reform [61], and its curriculum system aligns closely with that of other core provinces, such as Guangdong and Jiangsu [62]. Consequently, the entrepreneurship education curriculum in Zhejiang Province is highly representative. Secondly, the entrepreneurial environment in Zhejiang Province is typical. As China's most active province for the private economy, it offers a typical context for entrepreneurship education research. Finally, the institutions are selected using a stratified control approach. The three chosen institutions comprise a "Double First Class" comprehensive university, a regular undergraduate institution, and a vocational college, reflecting the elite, applied, and skill levels of China's higher education system [63,64]. This stratified sampling design enhances the representativeness of the sample in relation to the national structure of higher education institutions.

This study distributed questionnaires to students after obtaining official and student consent by contacting the university's entrepreneurship guidance departments. The online survey link was randomly distributed over the internet to college students. Out of the 1,350 questionnaires sent, 1,013 valid responses were received, yielding a response rate of 75.04%. To reduce potential bias—for example, non-respondents may hold different opinions or characteristics—the study adopted random distribution and anonymous answers during the data collection process. Additionally, a preliminary examination of the non-respondent sample found that the differences between non-respondents and valid respondents in key variables such as gender, major, and whether family members have entrepreneurial experience were small, further reducing the potential risk of bias.

The study's participants were college students. The majority of the individuals in the valid sample were aged between 18 and 22. The sample consisted of 203 males and 810 females. Of the students, 372 had family members with entrepreneurial experience, whereas 641 did not. Specifically, 80 male students had entrepreneurial experience within their families, while 123 did not; 292 female students had entrepreneurial experience among their family members, and 518 did not. Furthermore, 360 students were only children, as opposed to 653 who were not; 454 participants majored in natural sciences, and 559 in humanities and social sciences. The study has been approved by the research ethics committee of the researcher's institution. Informed consent was obtained from all participants before data collection began. They were informed that their participation was entirely voluntary and that they could withdraw from the survey at any point.

## 3.3 Measure

**3.3.1 Entrepreneurial Intention Scale.** This research employed the Individual Entrepreneurial Intention Scale, initially developed by Liñán and Chen in 2009 and subsequently revised by Jena in 2020 [65,66]. The scale has been translated into Chinese and implemented in Chinese universities [48]. It utilizes a five-point Likert scale, where 1 signifies "strongly disagree" and 5 signifies "strongly agree." Higher scores reflect a stronger entrepreneurial intention (EI) among participants. The questionnaire consists of six items, such as "If I have the opportunity and resources, I want to start a business" and "Given various options, I prefer to start a business." The overall Cronbach's α for the scale was 0.954, indicating its excellent reliability.

**3.3.2 Career adaptability scale.** This research employed the Career Adaptability Scale, developed by Savickas and Porfeli in 2012 [67]. The scale, previously translated into Chinese and utilized within Chinese university settings, was also implemented in this study [68]. The scale consists of 24 items, which are distributed across four dimensions: career

concern, career control, career curiosity, and career confidence. Sample questions include "I will think about what my future holds" and "I will prepare for my future." The scale utilizes the Likert five-point scale format. Participants rated the items on a scale ranging from 1 (strongly disagree) to 5 (strongly agree). The overall Cronbach's α for the scale was 0.974, signifying its excellent reliability.

**3.3.3 Entrepreneurial self-efficacy scale.** This study utilized the Entrepreneurial Self-Efficacy Scale developed by Wilson et al. (2007) [69]. Participants were requested to assess their entrepreneurial abilities relative to those of their peers. Responses were recorded on a 5-point Likert scale, ranging from "significantly worse" to "significantly better than peers." Higher scores on this scale indicate stronger perceptions of entrepreneurial self-efficacy (ESE). The scale has also been translated into Chinese and implemented in Chinese universities [30]. The overall Cronbach's α for the scale was 0.948, signifying excellent reliability.

**3.3.4 Entrepreneurial passion scale.** This study employed the Entrepreneurial Passion Scale developed by Cardon et al. (2013), which is specifically designed for the invention and discovery phases of entrepreneurial projects [38]. The scale is unidimensional and comprises five items. It utilizes a five-point Likert scale, where 1 signifies "strongly disagree" and 5 signifies "strongly agree." An example item is, "I am motivated to think about how to improve existing products/ services." This scale has been validated as a reliable and valid measure of entrepreneurial passion among Chinese individuals [70]. The overall Cronbach's α for the scale was 0.964, indicating excellent reliability.

**3.3.5 Control variables.** This study incorporates three control variables: gender, major, and whether family members have entrepreneurial experience. These variables have been widely considered in studies related to EI [59,71].

## 3.4 Data analysis

This study utilized Confirmatory Factor Analysis (CFA) with a multi-factor model to evaluate common method variance (CMV), comparing it to a single-factor model [72]. The multi-factor model showed a good fit to the data ($\chi^2 = 3072.396$, df = 758), while the single-factor model displayed a poor fit ($\chi^2 = 19100.45$, df = 779). The comparison indicated that the multi-factor model significantly outperformed the single-factor model ($\Delta\chi^2 = 16028.054$, $\Delta df = 21$, p < 0.001). This suggests that the multi-factor model was substantially superior, indicating that the issue of CMV in this study was minimal [72].

After controlling for common method variance, descriptive analysis, correlation analysis, and model testing were conducted on the data in accordance with the study hypotheses. Initially, descriptive statistics were performed on the variables to evaluate data concentration and dispersion. Subsequently, the chain mediation effects were examined using Pearson correlation analysis and the PROCESS plug-in for SPSS. The PROCESS plug-in is utilized in conjunction with moderating and mediation analyses, based on path analysis [73]. This study employs Model 6 of the PROCESS macro plug-in for SPSS as the analytical framework, with career adaptability (CA) designated as the independent variable, entrepreneurial passion (EP) and entrepreneurial self-efficacy (ESE) as chain mediating variables, and entrepreneurial intention (EI) as the dependent variable. The selection of Model 6 is due to its ability to test the chain effects of multiple mediating variables and systematically analyze the multi-path indirect effects of CA on EI through EP and ESE [73]. In contrast, alternative models, (such as Model 4 or Model 7, cannot simultaneously capture the relationship between multiple mediating variables and are thus unsuitable for hypothesis testing in this study [73]. Additionally, most current empirical studies on chain mediation models have adopted this data analysis method [74,75].

## 4. Results

### 4.1 Descriptive statistics and correlation analysis

Normality analyses were conducted on all continuous variables. The absolute values of skewness for each item ranged from −0.324 to 0.253, and the absolute values of kurtosis ranged from −0.361 to 0.116. The results met the criteria of skewness being less than 2 and kurtosis being less than 7 in absolute value, indicating that the data follows a normal distribution [76]. Therefore, the prerequisites for subsequent data analysis were satisfied [77]. The data from the 1013

validated questionnaire scores obtained were subjected to descriptive statistics to derive the mean and standard deviation of the total scores on the three scales of CA, ESE, EP, and EI, respectively. The mean value of the total score for CA was 3.59 (SD = 0.672), for ESE was 3.2 (SD = 0.807), for EP was 3.284 (SD = 0.839), and for EI was 3.079 (SD = 0.87). Since all four scales use a five-point rating system, the data revealed that college students' CA, ESE, EP, and EI were generally at moderate to high levels.

The means, standard deviations, and correlation matrices for each variable in the study are presented in Table 1. The results indicated that CA, ESE, EP, and EI were previously significantly correlated with each other, with correlation coefficients ranging from 0.543 to 0.748. This also provided initial support for the subsequent regression analysis.

## 4.2 Difference analysis

This study employed t-test to examine the differences in CA, EP, ESE, and EI based on gender, major, and family members' entrepreneurial experience. The results indicated significant differences in CA (t = 2.154; p < 0.05), EP (t = 2.871; p < 0.01), ESE (t = 4.974; p < 0.001), and EI (t = 4.926; p < 0.001) based on gender, with males scoring higher than females. No significant differences in CA (t = 1.783; p > 0.05) were found based on major, but significant differences were observed in EP (t = 2.024; p < 0.05), ESE (t = 2.078; p < 0.05), and EI (t = 2.302; p < 0.05), with students in natural sciences scoring higher than those in humanities and social sciences. Additionally, significant differences in CA (t = 6.132; p < 0.001), EP (t = 5.584; p < 0.001), ESE (t = 5.751; p < 0.001), and EI (t = 5.86; p < 0.001) were found based on whether family members had entrepreneurial experience, with students whose family members had entrepreneurial experience scoring higher than those whose family members did not.

## 4.3 Mediating effect test

Model 6 was selected for analysis using the PROCESS plug-in for SPSS. The model was tested with CA as the independent variable, EP and ESE as chain mediating variables, and EI as the dependent variable, yielding the data results in Table 2. As shown in Table 2, CA positively and significantly predicted EI ($\beta$ = 0.516, p < 0.001), thus supporting hypothesis 1. When EP and ESE were introduced as mediating variables, the impact of CA on EI was not significant ($\beta$ = −0.051, p > 0.05), but CA had a significant positive effect on both EP ($\beta$ = 0.635, p < 0.001) and ESE ($\beta$ = 0.477, p < 0.001). EP had a significant impact on ESE ($\beta$ = 0.31, p < 0.001) and EI ($\beta$ = 0.583, p < 0.001), and ESE also had a significant positive impact on EI ($\beta$ = 0.294, p < 0.001).

The mediation effect test results for EP indicate that the 95% confidence interval's upper and lower limits do not include zero, suggesting that EP mediates the relationship between CA and EI, with a mediation effect of 0.479, which constitutes 71.6% of the total effect. The indirect effect via the CA-EP-EI pathway accounts for the majority of the total effect (71.6%), demonstrating that entrepreneurial passion significantly mediates the influence of career adaptability on entrepreneurial intention. Hypothesis 2 is supported. The mediation effect test results for ESE reveal that ESE also mediates the impact of CA on EI, with a mediation effect of 0.181, constituting 27.1% of the total effect. The indirect effect via the CA-ESE-EI

**Table 1. Summary table of descriptive statistics and correlation analysis.**

| Variable | M | SD | 1 | 2 | 3 | 4 |
|---|---|---|---|---|---|---|
| CA | 3.59 | 0.672 | **0.728** | | | |
| ESE | 3.2 | 0.807 | 0.696*** | **0.891** | | |
| EP | 3.284 | 0.839 | 0.652*** | 0.64*** | **0.935** | |
| EI | 3.079 | 0.87 | 0.543*** | 0.654*** | 0.748*** | **0.903** |

Note. Bolded fonts are AVE root values; *** p < .001.

pathway accounts for 27.1% of the total effect, illustrating that entrepreneurial self-efficacy plays a mediating role in the influence of career adaptability on entrepreneurial intention. Hypothesis 3 is supported. The test results for the chain mediation effect of EP and ESE show that the serial mediation effect is significant, with a mediation effect of 0.075, constituting 11.2% of the total effect. The indirect effect via the CA-EP-ESE-EI pathway accounts for 11.2% of the total effect, indicating that entrepreneurial passion and entrepreneurial self-efficacy play a chain mediation role in the influence of career adaptability on entrepreneurial intention (see Table 3). Hypothesis 4 is supported. Given the significant total effect of the aforementioned chain mediation model ($\beta = 0.669$, 95% CI [0.000, 0.601]), and the non-significant direct effect ($\beta = -0.066$, 95% CI [−0.142, 0.84]), the significant indirect effect explains all the variance, suggesting a complete mediation effect [73]. Numerous academic studies have also utilized this determination method and standard [78,79]. Therefore, it can be

**Table 2. Regression analysis of the relationship between variables.**

| Regression equation | | Overall fit index | | Significance of regression coefficients | | |
|---|---|---|---|---|---|---|
| **Result Variables** | **Predictive variables** | **R2** | **F** | **β** | **SE** | **t** |
| EI | CA | 0.314 | 115.55*** | 0.516 | 0.035 | 19.334*** |
| | Gender(female=0) | | | 0.102 | 0.057 | 3.896*** |
| | Major(HSS=0) | | | 0.053 | 0.046 | 2.026* |
| | EEFM/NEEFM(NEEFM=0) | | | 0.085 | 0.048 | 3.172** |
| EP | CA | 0.433 | 192.088*** | 0.635 | 0.03 | 26.126*** |
| | Gender(female=0) | | | 0.047 | 0.05 | 1.965* |
| | Major(HSS=0) | | | 0.035 | 0.04 | 1.479 |
| | EEFM/NEEFM(NEEFM=0) | | | 0.063 | 0.042 | 2.601** |
| ESE | CA | 0.557 | 253.713*** | 0.477 | 0.033 | 17.149*** |
| | EP | | | 0.31 | 0.037 | 11.136*** |
| | Gender(female=0) | | | 0.104 | 0.043 | 4.935*** |
| | Major(HSS=0) | | | 0.04 | 0.034 | 1.871 |
| | EEFM/NEEFM(NEEFM=0) | | | 0.035 | 0.036 | 1.614 |
| EI | CA | 0.61 | 262.392*** | −0.051 | 0.038 | −1.729 |
| | EP | | | 0.583 | 0.029 | 21.045*** |
| | ESE | | | 0.294 | 0.032 | 9.918*** |
| | Gender(female=0) | | | 0.04 | 0.044 | 1.933* |
| | Major(HSS=0) | | | 0.018 | 0.035 | 0.371 |
| | EEFM/NEEFM(NEEFM=0) | | | 0.032 | 0.037 | 0.115 |

Note: β means standardized coefficient; *p<0.05; **p<0.01; ***p<0.001; EEFM means entrepreneurial experience of family members; NEEFM means no entrepreneurial experience of family members; HSS means major in Humanities and social sciences.

**Table 3. The direct, indirect, and total effect of chain mediation model.**

| | Path | Effect | Percentage of total effect | Lower limit of 95% CI | Upper limit of 95% CI |
|---|---|---|---|---|---|
| Total effect | CA→EI | 0.669 | – | 0 | 0.601 |
| Direct effect | CA→EI | −0.066 | −9.9% | −0.142 | 0.84 |
| Indirect effect | CA→EP→EI | 0.479 | 71.6% | 0.402 | 0.555 |
| | CA→ESE→EI | 0.181 | 27.1% | 0.123 | 0.244 |
| | CA→EP→ESE→EI | 0.075 | 11.2% | 0.046 | 0.111 |

Note: *p<0.05; **p<0.01; ***p<0.001.

regarded as a fully mediated model. This implies that the impact of CA on EI is fully mediated through EP and ESE (see Fig 2).

To evaluate the potential impact of gender bias, this study incorporated gender as a covariate in the regression model. This adjustment resulted in only minor alterations to the path coefficients. For instance, the total effect of career adaptability on entrepreneurial intention shifted from $\beta = 0.524$ to $\beta = 0.516$, a change of $\Delta\beta = 0.8\%$, suggesting that gender did not significantly affect the relationships among the core variables. Furthermore, gender group comparisons were conducted using 5000 Bootstrap samples, with the outcomes presented in Table 4. The results are as follows: CA→EP: $\Delta\beta = 0.006$, $p > 0.05$; CA→ESE: $\Delta\beta = 0.049$, $p > 0.05$; EP→ESE: $\Delta\beta = 0.081$, $p > 0.05$; EP→EI: $\Delta\beta = 0.027$, $p > 0.05$; ESE→EI: $\Delta\beta = 0.024$, $p > 0.05$. The data indicates that there are no significant differences in the key paths.

## 5. Discussion and implication

Although previous studies have indicated that career adaptability (CA) has a positive effect on entrepreneurial intention (EI), research in the context of East Asia has been scarce [25,31]. Importantly, little attention has been given to exploring the relationship between career adaptability and entrepreneurial intention from the perspective of career development, particularly by integrating cognitive and emotional factors. This study aims to expand on the potential influencing factors of the relationship between CA and EI within the context of China. The empirical research was conducted with college students from universities designated by the Ministry of Education of China as demonstration schools for entrepreneurship education. These students have all participated in a semester-long course on "Career Planning," making them a group highly susceptible to the influence of CA. The main objective of this study is to validate the theoretical pathway of 'career adaptability → entrepreneurial passion →entrepreneurial self-efficacy → entrepreneurial intention,' rather than to make cross-regional group inferences. Therefore, selecting three universities in Zhejiang Province with a homogeneous policy

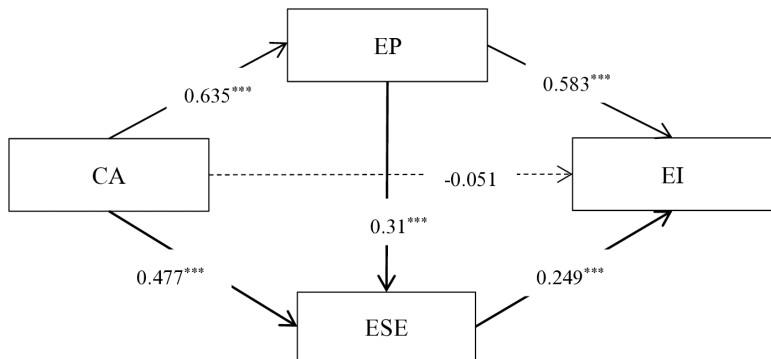

**Fig 2. Model diagram of the effect of CA on EI.**

**Table 4. Intergroup difference testing of gender in the main path of the model.**

| Path | Female Standardized Regression Coefficients | Male Standardized Regression Coefficients | Significance of Group Difference |
|---|---|---|---|
| CA→EP | 0.812 | 0.806 | $\Delta\beta = 0.006$, $p = 0.901$ |
| CA→ESE | 0.561 | 0.610 | $\Delta\beta = 0.049$, $p = 0.423$ |
| EP→ESE | 0.286 | 0.367 | $\Delta\beta = 0.081$, $p = 0.067$ |
| EP→EI | 0.601 | 0.628 | $\Delta\beta = 0.027$, $p = 0.682$ |
| ESE→EI | 0.314 | 0.338 | $\Delta\beta = 0.024$, $p = 0.725$ |

environment (all recognized as model schools for innovation and entrepreneurship by the Ministry of Education) helps to reduce the confounding effects caused by differences in policy environments and enhances internal validity [80]. This strategy has been widely adopted in entrepreneurship education research [81,82]. This study focused on the impact of CA on EI and the potential mechanisms involved. Through a chain mediation model, it was found that CA influenced EI through three indirect pathways: entrepreneurial self-efficacy (ESE) fully mediated the relationship between CA and EI; entrepreneurial passion (EP) fully mediated the relationship between CA and EI; and both ESE and EP served as chain mediators between CA and EI.

## 5.1 Theoretical implications

First, the results of this study confirmed Hypothesis 1, indicating that college students' CA significantly and positively influences EI. This was consistent with previous research findings that individuals with higher CA typically exhibit greater EI [25,31]. This result may stem from higher CA enabling individuals to better cope with the ever-changing market environment, thereby increasing the likelihood of developing EI [25,31]. However, further multilevel regression analysis indicated that when entrepreneurial passion (EP) and entrepreneurial self-efficacy (ESE) are included, CA does not directly predict EI. This finding aligned with results from a study conducted on job seekers in Togo [16]. The reason can be explained by the Social Cognitive Theory (SCT), which posits that the environment, the individual, and behavior mutually influence each other [18]. Specifically, while CA as a personal factor shaped by trait–environment interaction helps individuals adapt to the ever-changing market environment, the formation of EI is also influenced by other key personal factors, such as EP (positive emotions towards entrepreneurship and identity recognition) and ESE (confidence in entrepreneurial abilities) [18]. The comprehensive evaluation of these personal factors determines individuals' willingness to pursue entrepreneurial activities. As some studies suggest, perceiving significant entrepreneurial barriers might lead to frustration, potentially resulting in a willingness to seek paid employment [83–85]. Although individuals with high career adaptability could adjust to a changing market environment, they might also feel frustrated due to perceived entrepreneurial barriers [84], which could reduce their entrepreneurial passion (EP) and entrepreneurial self-efficacy (ESE), temporarily suppressing their entrepreneurial intentions (EI) and leading them to choose paid employment instead [85]. Therefore, EP and ESE play an important mediating role in the relationship between career adaptability (CA) and EI. If an individual's levels of EP and ESE are low, CA may not have a significant impact on EI. The study found that CA only indirectly affects college students' EI through the chain mediation of EP and ESE. This finding also suggests that among environmental, cognitive, and emotional factors, cognitive and emotional aspects may be more direct predictors of EI. In other words, in the process of CA influencing EI, EP and ESE may play a more critical role. Thus, when studying the impact of CA on EI, it is necessary to comprehensively consider cognitive and emotional factors. This includes not only recognizing the independent effects of these factors but also emphasizing their dual-system structure in processing characteristics. The results of this study indicate that CA, as a capacity to adapt to professional environments, has varying degrees of influence on students' EP, ESE, and EI. This finding suggests that higher education administrators should not only focus on enhancing students' CA but also effectively improve their EP and ESE, which may lead to an increase in EI.

Secondly, the results of this study indicated that EP played a mediating role in the relationship between CA and EI, confirming Hypothesis 2. This finding was consistent with the results of research conducted by Liang and Wang (2016) in China [37]. Their study examined these three variables and emphasized the important role of CA in enhancing EP, while EP was identified as a key factor influencing EI [37]. Specifically, university students with higher levels of CA are better able to cope with entrepreneurial challenges, experience positive emotions, and strengthen their identification with the entrepreneurial role. This positive emotion and identification ultimately translate into a strong EI [54]. This phenomenon can be explained by the Broaden-and-Build Theory of Positive Emotions (BBTPE) [34]. According to this theory, EP not only expands an individual's resource capabilities but also enhances cognitive flexibility, broadens cognitive scope,

thereby promoting the development of resource knowledge structures and improving entrepreneurial capabilities and intentions [34].

Thirdly, this study found that ESE mediates the relationship between CA and EI, thereby supporting Hypothesis 3. This result is consistent with previous studies conducted on Serbian university students [25] and Chinese university students [59]. Their studys also explored the relationships among these three variables, confirming that CA is an important factor in enhancing ESE, while ESE is a key factor influencing EI. This outcome may be due to individuals with higher CA typically exhibiting greater confidence when formulating business plans, analyzing business opportunities, and setting goals, which increases their tendency to develop EI. SCT suggests that individuals with high self-efficacy are able to set higher goals, create more effective plans, maintain sustained effort, actively utilize feedback, and resiliently overcome setbacks [18]. These characteristics are particularly important in entrepreneurship, which often involves ambiguity, risk, and instability; thus, effort, perseverance, and adaptability become crucial [28]. Therefore, individuals with higher CA generally demonstrate stronger confidence in the entrepreneurial process, thereby enhancing their EI.

Finally, the results of this study indicate that EP and ESE play a chain mediating role between CA and EI, thereby validating Hypothesis 4. Chain mediation is defined as "a process in which an independent variable sequentially affects the dependent variable via two or more consecutive mediators" [86]. Although previous research has suggested that CA may influence EI through ESE [16], this study highlights the significance of the chain mediating effect, adding to the understanding of the role of emotional factors (EP) in promoting entrepreneurial confidence and intention. This study further clarifies the pathway through which CA indirectly affects EI via EP and ESE, emphasizing the interaction between emotional and cognitive factors. This suggests that the impact of CA on EI is not only linear but also involves a multidimensional emotional-cognitive linkage mechanism. This finding can be explained using SCT. CA, as a personal factor of trait-environment interaction, influences EI (behavioral factor) through the mediating effects of EP (emotional factor) and ESE (cognitive factor). The positive impact of entrepreneurial passion (EP) on entrepreneurial self-efficacy (ESE) may be attributed to the way emotions influence individuals' perceptions and their approaches to understanding situations [87]. When individuals are passionate about entrepreneurship, they are more inclined to seek out and develop skills pertinent to entrepreneurial activities, thereby enhancing their execution capabilities and increasing their ESE [87]. This phenomenon can also be explained by the Broaden-and-Build Theory of Positive Emotions (BBTPE). BBTPE suggests that EP, as an emotional resource that activates self-efficacy, can expand individuals' engagement and enjoyment in entrepreneurial activities, thereby enhancing their entrepreneurial capabilities and confidence [88].Therefore, individuals with higher CA may better adapt to market changes, gain positive experiences from entrepreneurial activities, enhance EP, and subsequently improve their entrepreneurial capabilities and information, thereby influencing college students' EI.

Furthermore, this study is conducted within the context of China, where cultural factors exert a certain influence on the research conclusions. China is deeply influenced by Confucian values, with a prevailing collectivist culture, and these cultural traits have a profound effect on individuals' career choices and entrepreneurial intentions [89,90]. The concepts emphasized by Confucian thought, such as the "Doctrine of the Mean" and "contentment with the status quo" [91], may lead individuals to be more cautious when facing entrepreneurial risks. If they lack entrepreneurial passion and a sense of entrepreneurial self-efficacy, individuals may be more inclined to choose stable career paths, thereby suppressing the formation of entrepreneurial intentions. However, the sense of responsibility inherent in Confucian culture, encapsulated by the idea of "cultivating oneself, regulating the family, governing the state, and bringing peace to the world" [91], may also inspire some individuals to realize personal value and contribute to society through entrepreneurship, thereby enhancing their entrepreneurial passion and self-efficacy. Consequently, Chinese college students, influenced by Confucian values, perceive that entrepreneurial passion and self-efficacy play a more critical role in the relationship between career adaptability and entrepreneurial intentions. In a collectivist cultural context, individual decision-making is often constrained by family and social-group expectations [92]. If family members possess entrepreneurial experience, they may enhance an individual's career adaptability, entrepreneurial passion, and self-efficacy through mentorship and resource support;

conversely, individuals without such family backgrounds may be at a disadvantage in forming entrepreneurial intentions due to a lack of resources and experience. Thus, family and social group expectations may also play a moderating role in the process of career adaptability influencing entrepreneurial intentions. Since this study primarily focuses on the theoretical pathway of CA-EP-ESE-EI, the presence or absence of entrepreneurial experience among family members is controlled as a variable. Unlike in China, in the West, individualism primarily stimulates entrepreneurial passion through personal achievement, with individuals being less influenced by external collective expectations. This may lead to differences in the strength of the mediating effects of entrepreneurial passion and entrepreneurial self-efficacy within the CA-EP-ESE-EI theoretical pathway.

In summary, this study investigates the factors that influence entrepreneurial intention (EI) from the perspective of career development, offering empirical evidence and practical insights into enhance EI. Theoretically, it establishes a chain mediation model to examine the intricate relationships and conditions among career adaptability (CA), entrepreneurial passion (EP), entrepreneurial self-efficacy (ESE), and EI. Previous researches on entrepreneurial intention have largely relied on the Theory of Planned Behavior (TPB) [93,94]. In contrast, this study incorporates Social Cognitive Theory (SCT) and the Broaden-and-Build Theory of Positive Emotions (BBTPE) to provide a more comprehensive framework for understanding the interplay between CA, EP, ESE, and EI. While TPB highlights the predictive roles of attitudes, subjective norms, and perceived behavioral control on intentions, it falls short of explaining the dynamic interplay between environmental and personal factors. SCT, however, offers a more thorough exploration of the complex mechanisms behind behavior generation by considering the interactions among environmental, emotional, and cognitive factors. The results of this study not only expand the research perspectives on CA and EI but also provide empirical support for SCT and BBTPE in the entrepreneurial domain. Although the research sample is confined to a specific province, the model derived from SCT has been validated in cross-cultural entrepreneurship studies [18,25,95]. For example, the influence of career adaptability on entrepreneurial intention through self-efficacy has been significant in a Serbian sample from Europe [25], and the impact of career adaptability on entrepreneurial intention is also significant in a Ghanaian sample from Africa [95]. Consequently, this model can offer a transferable analytical framework for other nations with comparable entrepreneurial education policies and cultural contexts, such as those in East Asia and Southeast Asia.

## 5.2 Practical implications

This study has broad practical significance. The results provide valuable guidance for higher education institutions on how to promote students' EI. The findings indicated that EP and ESE played a crucial mediating role in the transformation of CA into EI. Therefore, universities should not only offer courses such as "Career Planning," but also introduce more entrepreneurship-oriented courses (such as "Entrepreneurship Courses") to support the development of EP, ESE, and EI. Specific measures can be implemented in the following ways. First, universities can make some improvements to the existing "Career Planning" course. This course can incorporate modules that stimulate entrepreneurial passion and cultivate a sense of entrepreneurial self-efficacy. For example, case discussions based on real entrepreneurial situations can be included, or simulated entrepreneurial activities can be conducted to enhance students' emotional identification and confidence in entrepreneurship through experience. Given that EP accounts for the vast majority of the total effect, this finding highlights the critical role of emotional factors (such as EP) in entrepreneurship education. Universities can focus on content that promotes the development of EP to facilitate the transformation from CA to EI. For instance, inviting entrepreneurs to share their success stories or teachers to showcase their acknowledgment of the entrepreneurial spirit and express their own entrepreneurial enthusiasm can inspire students. In contrast to Western individualism, which sparks passion through personal accomplishments [33], Chinese students' entrepreneurial passion (EP) is also shaped by the social evaluation system. When invited entrepreneurial role models are given institutional recognition (such as being honored with the title "Model Entrepreneur" by the government), their narratives resonate more deeply with the Confucian ideal of "cultivating oneself, regulating the family, governing the state, and bringing peace to the world," thereby more

effectively stimulating students' EP [91]. Secondly, universities can bolster the development of entrepreneurship courses. "Entrepreneurship Courses" can encompass topics such as project management, innovative thinking, market analysis, and resource acquisition, while also integrating group projects and entrepreneurship competitions to foster students' practical skills. By evaluating individual and team performance in entrepreneurial projects, students can gain confidence in their own capabilities. Finally, enhancing practical experience is crucial. Strengthening entrepreneurship competitions and practical experiences can further enhance EP and ESE. For example, organizing field trips to companies or tours of entrepreneurial incubators can offer practical opportunities for students to engage directly with the entrepreneurial environment. This not only ignites enthusiasm but also boosts students' ESE through real-world tasks. The research also indicates that female students have lower levels of EP and ESE compared to male students, suggesting that entrepreneurial interventions aimed at female college students can optimize outcomes by boosting EP and ESE, such as hosting workshops specifically designed to increase entrepreneurial confidence. Students with family members who have entrepreneurial experience tend to score higher in CA, EP, and ESE, implying that the influence of entrepreneurial families may reinforce adaptability, enthusiasm, and confidence. For students without such backgrounds, additional supportive programs can be created to address this disparity.

Additionally, universities can use tools or strategies to measure improvements in EP and ESE to assess the impact of promoting CA on EI. For instance, throughout the implementation process, standardized scales (e.g., entrepreneurial passion and self-efficacy scales) can be employed for pre- and post-assessments of students. This practice enables monitoring of the long-term effects of educational interventions on entrepreneurial passion (EP) and entrepreneurial self-efficacy (ESE). The creation of online learning platforms to document data on students' engagement in entrepreneurship-related activities, such as completed entrepreneurial projects and competition outcomes, can facilitate dynamic tracking of their growth trajectory. By conducting longitudinal follow-ups on students' transitions from university to employment, the influence of career adaptability (CA) enhancement on career path choices—including those that are not entrepreneurial—can be assessed.

The findings of this study also have important implications for government and education policymakers. The education sector can formulate policies to integrate entrepreneurship education into the compulsory curriculum of higher education institutions, while also encouraging universities to develop interdisciplinary courses related to CA, EP, and ESE. This will ensure that a greater number of students have access to entrepreneurship education resources. The government can support entrepreneurship education and competition projects through special funding, providing resources for universities to develop entrepreneurship courses, establish entrepreneurship laboratories, or subsidize students' participation in entrepreneurial practices. Supportive policies can be established, such as tax reductions for successful entrepreneurial graduates from universities or rewards for institutions that excel in entrepreneurship education, to motivate more educational institutions to prioritize entrepreneurship education.

Furthermore, enhancing one's career adaptability (CA) may also foster career outcomes in non-entrepreneurial sectors. For example, increased CA could assist individuals in adapting more effectively to career transitions, planning their career development, and improving their employability in a volatile job market. Therefore, educational interventions targeting CA can not only be used to improve EI but can also be widely applied in vocational education and career guidance, providing support for students' overall career development.

## 6. Limitations and future research directions

This study is not without its limitations. Firstly, it employs a cross-sectional design; therefore, future research is encouraged to utilize longitudinal designs or experimental methods to verify the identified causal relationships. Longitudinal studies could provide time-series data by tracking changes in entrepreneurial passion and entrepreneurial self-efficacy among the same cohort of students. For instance, such studies could explore whether students experience an increase in entrepreneurial passion or entrepreneurial self-efficacy during the process of developing entrepreneurial intentions after

receiving entrepreneurship education. Experimental methods, on the other hand, could clarify the causal effects of specific interventions (e.g., entrepreneurship courses or supervisors) on students' entrepreneurial intentions by controlling for variables. For example, a randomized controlled trial could be designed, where one group of students receives entrepreneurship education interventions while the other does not, and the differences in entrepreneurial intentions between the two groups are compared.

Secondly, the geographical limitations of the sample necessitate caution in generalizing the conclusions. This study investigated several demonstration universities in Zhejiang Province. Although the sample from Zhejiang Province is representative of policy, its economic activity level may be higher than the national average, and the regional selection of the sample may restrict the broader applicability of the research. Therefore, future studies could expand the survey scope to other regions to test the universality of the findings. For example, validating the model's applicability in underdeveloped areas; additionally, future research could also include different types of institutions, incorporating private universities and vocational colleges to enhance the sample's representativeness across various types of higher education institutions.

This study concentrates on the entrepreneurial intentions of Chinese students, which may, to some extent, reflect China's unique cultural and policy context. Future research could enhance the international applicability of the research model by conducting comparative studies across countries with significantly different cultural and economic backgrounds, as well as variations in entrepreneurial education policies. For instance, the government-led model of entrepreneurial education in China differs from the market-driven model in some developed countries in Europe and America [96]. Subsequent studies could compare the entrepreneurial intentions of university students in different institutional environments. Future research should consider the impact of different cultures on entrepreneurial education and intentions. For example, in countries where entrepreneurial education is not as emphasized, do students exhibit similar entrepreneurial passion and intentions as Chinese students? Additionally, comparative studies in different cultural contexts could explore the universality and cultural specificity of entrepreneurial education.

In addition, this study has a gender imbalance in the sample (80% female). Although data analysis indicates that the core mechanisms have cross-gender stability, future research still needs to validate these pathways in balanced samples, especially in certain cultural contexts where the development of entrepreneurial self-efficacy may exhibit gender differences [97].

Lastly, while this study primarily focused on the impact of CA on EI, future research could explore other potential influencing factors to uncover more complex mechanisms. Examples include personality traits, external factors (such as access to funding opportunities or support from entrepreneurial ecosystems), and different forms of entrepreneurship education. Furthermore, as this study concentrated on university students, future studies could extend to employees in various industries to examine how CA influences EI in different occupational contexts.

## 7. Conclusion

This study reveals the fundamental mechanisms behind the development of entrepreneurial intention (EI) among university students, providing fresh insights for both the theory and practice of entrepreneurship education. The results indicate that while career adaptability (CA) is significantly and positively correlated with EI, its effect is indirect, mediated by entrepreneurial passion (EP) and entrepreneurial self-efficacy (ESE). Specifically, the influence of CA on EI is transmitted through three critical mediating pathways: the complete mediating effect of EP, the complete mediating effect of ESE, and the sequential mediating effect of EP followed by ESE. These findings suggest that the career adaptability of university students should be directed through enhancements in emotional drivers and psychological beliefs to effectively convert into EI.

These findings have significant implications for optimizing entrepreneurship education. Traditional entrepreneurship education often emphasizes the development of students' entrepreneurial knowledge and skills. However, this study highlights the crucial role of emotional and cognitive factors in the formation of entrepreneurial intention. In the future,

entrepreneurship education should transition to a more multidimensional and integrated development model that not only enhances students' abilities but also fosters their entrepreneurial passion and strengthens their entrepreneurial self-efficacy. Such a transformation would not only better support students in achieving their entrepreneurial goals but also contribute to the broader promotion and development of an entrepreneurial culture.

## Author contributions

**Conceptualization:** Jiping Zhang.

**Data curation:** Jiping Zhang.

**Formal analysis:** Jiping Zhang.

**Funding acquisition:** Jiping Zhang.

**Investigation:** Jiping Zhang.

**Methodology:** Jiping Zhang.

**Project administration:** Jiping Zhang.

**Resources:** Jiping Zhang.

**Software:** Jiping Zhang.

**Supervision:** Jiping Zhang.

**Validation:** Jiping Zhang.

**Visualization:** Jiping Zhang.

**Writing – original draft:** Jiping Zhang.

**Writing – review & editing:** Jiping Zhang.

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
