## [Decision Letter · Decision Letter 0]

16 Jan 2025

Dear Dr. Zhang,

Thank you for submitting your manuscript to PLOS ONE. After careful consideration, we feel that it has merit but does not fully meet PLOS ONE’s publication criteria as it currently stands. Therefore, we invite you to submit a revised version of the manuscript that addresses the points raised during the review process.

We look forward to receiving your revised manuscript.

Kind regards,

Imran Anwar

Academic Editor

PLOS ONE

Journal Requirements:

Reviewers' comments:

Reviewer's Responses to Questions

**Comments to the Author**

1. Is the manuscript technically sound, and do the data support the conclusions?

Reviewer #1: Yes

Reviewer #2: Yes

2. Has the statistical analysis been performed appropriately and rigorously?

Reviewer #1: Yes

Reviewer #2: Yes

3. Have the authors made all data underlying the findings in their manuscript fully available?

Reviewer #1: Yes

Reviewer #2: Yes

4. Is the manuscript presented in an intelligible fashion and written in standard English?

Reviewer #1: Yes

Reviewer #2: Yes

Reviewer #1: Abstract

The abstract captures the essence of the research, summarizing the findings and implications succinctly.

Suggestions:

The term "chain-mediated effects" might be unfamiliar to some readers. Consider briefly defining it in lay terms.

The phrase "significantly positively correlated" is repeated. Instead, emphasize the practical significance or contribution of the findings.

Include a sentence highlighting the broader implications for entrepreneurship education globally, beyond the Chinese context.

Introduction

The introduction frames the research problem well, linking entrepreneurial intention (EI) to career adaptability (CA) using both theoretical and practical justifications. The rationale for including entrepreneurial passion (EP) and entrepreneurial self-efficacy (ESE) as mediators is strong and supported by relevant literature.

Suggestions:

Discuss the global relevance of CA in entrepreneurship. Does CA influence EI differently across cultural or economic contexts? This could broaden the appeal of your research.

Strengthen the argument about why previous studies on the direct CA-to-EI link yielded inconsistent results. Delve into potential gaps in methodology or theoretical frameworks.

Expand on how the study contributes to resolving these inconsistencies, particularly through the novel use of Social Cognitive Theory (SCT) and chain mediation.

Methodology

The selection of well-validated scales ensures reliability, and the large sample size enhances the robustness of the findings. Ethical considerations, including informed consent and institutional review board approval, are adequately addressed.

Suggestions:

Clarify the reasoning behind selecting only students from Zhejiang Province and the potential implications of this regional focus. Could this limit generalizability?

The high response rate (75.04%) is commendable, but discuss how potential biases (e.g., non-responders) were mitigated.

While Model 6 of the PROCESS macro is appropriate, explain why it was chosen over alternative models for testing mediation.

Results

Results are presented logically, with clear links to hypotheses. Tables and figures are used effectively to illustrate findings.

Suggestions:

Highlight the practical significance of the findings, beyond statistical significance. For instance, what does a 71.6% mediation effect of EP mean for entrepreneurship programs?

Discuss unexpected or non-significant results in more depth. For example, why was CA’s direct effect on EI non-significant, and how does this align with SCT?

Explore subgroup differences (e.g., gender, family entrepreneurial background) more thoroughly. Are there nuanced trends that could add depth to the analysis?

Discussion

The discussion effectively integrates findings with Social Cognitive Theory (SCT) and other theoretical frameworks, demonstrating a strong theoretical grounding. It highlights the mediating roles of EP and ESE as crucial pathways between CA and EI.

Suggestions:

Compare your findings with prior studies in more detail. How do the chain-mediated effects of EP and ESE in your study differ from or align with previous research?

Offer a critical evaluation of why SCT provides a better framework for understanding these relationships than other potential theories (e.g., Theory of Planned Behavior).

Discuss broader implications of the findings. For instance, could interventions to boost CA also enhance non-entrepreneurial career outcomes?

Implications

The practical suggestions for enhancing entrepreneurship education are timely and actionable.

Suggestions:

Provide specific examples of how universities can operationalize these insights. For instance, how can courses like "Career Life Planning" be modified to explicitly include elements that boost EP and ESE?

Discuss policy-level implications. How can governments or educational policymakers leverage your findings to improve national entrepreneurship outcomes?

Suggest tools or strategies for measuring improvements in EP and ESE over time, linking these to interventions aimed at boosting CA.

Limitations and Future Research

The study acknowledges key limitations, including its cross-sectional design and focus on university students in a specific region.

Suggestions:

The cross-sectional nature of the study is a significant limitation. Discuss how a longitudinal or experimental design could better establish causality.

Address the potential for cultural bias, given the focus on Chinese students. Would the findings hold in countries with less emphasis on entrepreneurship education?

Highlight the need for future research to examine other mediators (e.g., personality traits or external factors like access to funding) that could influence the CA-to-EI relationship.

Conclusion

The conclusion succinctly summarizes the findings and reiterates their importance.

Suggestions:

Avoid simply repeating the discussion points. Instead, use the conclusion to present a visionary outlook for the field. For example, how might this research transform the way entrepreneurship education is delivered?

Suggest a concrete next step for researchers or educators based on your findings.

Figures and Tables

Figures and tables are well-constructed and enhance comprehension.

Suggestions:

For Table 3, explicitly interpret the percentage contributions of direct and indirect effects to the total effect in the text.

References

The references are up-to-date and cover a broad range of relevant studies.

Suggestions:

Ensure consistency in citation formatting (e.g., journal names in italics).

Discuss any gaps in the literature that your study addresses explicitly.

These deeper comments aim to guide further refinement of the manuscript, ensuring both theoretical and practical contributions are maximized.

Reviewer #2: The present study is well structured and engaging still authors may increase its clarity and readability by explaining key terms and incorporating following suggestions.

1. Explain chain-mediated effect in the introduction and briefly distinguish this with mediation.

2. It is true that career adaptability as a variable and its link with EI are not explored much in the field of student entrepreneurship across the globe and studying only a region in this study may be biased. Author should either add it as a limitation or compare the results with other studies to present the study’s outcome more prominently.

3. High number of female students may also bring the bias in the study. Authors may also include the descriptive statistics of the data to make it more interesting. (eg. How many girls’ students are from business background family.)

4. Authors should check and explain if majority of the non-responders are from only one gender or any specific academic background.

5. Explain the reason behind the adoption of Pearson correlation analysis and the PROCESS plug-in for SPSS chain to examine the mediation effects over the other available alternatives.

6. Results are explained theoretically only in numbers. Kindly explain the interpretation and implication of these results especially non-significant in the light of social cognitive theory. Authors may add a “future research” section also and request the future researchers to explore the insignificant relation in other parts if the world.

7. Either in the discussion or in the introduction, Authors should add why they choose SCT over TPV or other alternative theories for this study.

**Do you want your identity to be public for this peer review?** For information about this choice, including consent withdrawal, please see our Privacy Policy

Reviewer #1: **Yes: ** Dr. Mushahid Ali Shamsi

Reviewer #2: **Yes: ** Alam Ahmad

---

## [Author Response · Author response to Decision Letter 1]

27 Jan 2025

Dear Editor:

It is our big honor to have you be our editor, and we appreciate your attention and recognition for our manuscript. Many thanks to reviewers for giving us a lot of valuable suggestions. Based on the suggestions of the reviewers, we have made a lot of revisions to the manuscript, and we are confident to submit a high-quality manuscript. Below are our responses based on the comments of the reviewers, thank you again.

To reviewers 1:

Dear reviewer 1:

Thank you very much for your time involved in reviewing the manuscript and your comments have further improved the quality of the manuscript. We have carefully reviewed the comments and revised the manuscript accordingly. Hope the explanation has fully addressed all of your concerns. Point-by-point responses to reviewers are as follows:

Q1:Abstract

The abstract captures the essence of the research, summarizing the findings and implications succinctly.

Suggestions:

The term "chain-mediated effects" might be unfamiliar to some readers. Consider briefly defining it in lay terms.

The phrase "significantly positively correlated" is repeated. Instead, emphasize the practical significance or contribution of the findings.

Include a sentence highlighting the broader implications for entrepreneurship education globally, beyond the Chinese context.

Reply1:Thank you for your valuable comments. We have rewritten the summary at your suggestion.

We replaced "significantly positively correlated" with "interrelated in ways that suggest meaningful pathways for enhancing entrepreneurial outcomes", avoiding repetition and emphasizing the value of the findings to the actual situation. We have also added a sentence of "While focused on Chinese university students, the implications extend globally..." to guide the significance of expanded research for global entrepreneurship education. For details, please see lines 48 - 70.

In addition, we also help the reader understand in the introduction by explaining that "chain-mediated" means that "a factor sequentially affects the final result through two or more continuous mediating variables", Please see for details lines 118-124. This idea is also reflected in the abstract, "This study examined how entrepreneurial self-efficacy and entrepreneurial passion jointly mediate the relationship between career adaptability and entrepreneurial intention." please see lines 49-51.

Q2:Introduction

The introduction frames the research problem well, linking entrepreneurial intention (EI) to career adaptability (CA) using both theoretical and practical justifications. The rationale for including entrepreneurial passion (EP) and entrepreneurial self-efficacy (ESE) as mediators is strong and supported by relevant literature.

Suggestions:

Discuss the global relevance of CA in entrepreneurship. Does CA influence EI differently across cultural or economic contexts? This could broaden the appeal of your research.

Strengthen the argument about why previous studies on the direct CA-to-EI link yielded inconsistent results. Delve into potential gaps in methodology or theoretical frameworks.

Expand on how the study contributes to resolving these inconsistencies, particularly through the novel use of Social Cognitive Theory (SCT) and chain mediation.

Reply2:Thank you for your valuable comments. We have rewritten the introduction, following your suggestion. The new version explores the global relevance of CA in entrepreneurship, see lines 87-91.

Results that explore the inconsistent direct effect of the existing literature CA-EI. The inconsistent results of studies on the relationship between CA and EI might be due to differences in theoretical frameworks, or neglect of key mediating variables [10], or different cultural or economic backgrounds [66]. For details, please see L93-95.

We explore the use of SCT as the theoretical basis, the key mediation variables and the chain mediation model, and the cultural background in China. Explore how to extend the CA-EI studies to address inconsistent arguments in previous studies. Please see lines 74-136 for further details. In addition, we have added some new literature.

Q3:Methodology

The selection of well-validated scales ensures reliability, and the large sample size enhances the robustness of the findings. Ethical considerations, including informed consent and institutional review board approval, are adequately addressed.

Suggestions:

Clarify the reasoning behind selecting only students from Zhejiang Province and the potential implications of this regional focus. Could this limit generalizability?

The high response rate (75.04%) is commendable, but discuss how potential biases (e.g., non-responders) were mitigated.

While Model 6 of the PROCESS macro is appropriate, explain why it was chosen over alternative models for testing mediation.

Reply:Thank you for your valuable comments. Your suggestions are of great help to improve the paper. Here are the modifications and explanations we have made, and we will reply one by one, as follows:

3.1 Clarify the reasoning behind selecting only students from Zhejiang Province and the potential implications of this regional focus. Could this limit generalizability?

Reply 3.1: This study mainly focuses on college students in Zhejiang Province, the reason is that Zhejiang Province, as a region with more developed entrepreneurship education, has a high level of economic development and a strong atmosphere of innovation and entrepreneurship. In addition, the three schools selected in this study were designated as vocational and entrepreneurship education model schools by the Ministry of Education of China, and these schools all offered "career planning" courses. All participants had completed the course and demonstrated high occupational resilience, which provided an ideal sample for the study (detailed in lines 300-306). However, sample selection by rationalization may impose limitations on the broad applicability of the study. Therefore, future studies could extend the survey to other regions, especially those with different cultural and economic backgrounds, to test the universality of the findings. This is detailed in limitations and future studies (see lines 683-700).

3.2 The high response rate (75.04%) is commendable, but discuss how potential biases (e.g., non-responders) were mitigated.

Reply 3.2: This study distributed the questionnaires to students after obtaining official and student consent by contacting the university's entrepreneurship guidance departments. The online survey link was distributed randomly over the internet to college students. Out of 1350 questionnaires sent, 1,013 valid responses were received, yielding a response rate of 75.04%. To reduce potential bias (for example, non-respondents may hold different opinions or characteristics), the study adopted random distribution and anonymous answers during the data collection process. In addition, a preliminary examination of the non-respondent sample found that the differences between non-respondents and valid respondents in key variables such as gender, major, and whether family members have entrepreneurial experience were small, further reducing the potential risk of bias (see lines 311-317).

3.3 While Model 6 of the PROCESS macro is appropriate, explain why it was chosen over alternative models for testing mediation.

Reply 3.3: We supplemented the article and also added some literature. Content is as follows:

Secondly, the chain mediation effects were examined using Pearson correlation analysis and the PROCESS plug-in for SPSS. The PROCESS plug-in is used with a combination of moderating and mediation analyses based on path analysis [57]. This study uses Model 6 of the PROCESS macro plug-in of SPSS as the analysis framework, with career adaptability (CA) set as the independent variable, entrepreneurial passion (EP) and entrepreneurial self-efficacy (ESE) set as chain mediating variables, and entrepreneurial intention (EI) as the dependent variable. The reason for choosing Model 6 is that this model can test the chain effects of multiple mediating variables and can systematically analyze the multi-path indirect effects of CA on EI through EP and ESE [57]. In contrast, other models (such as Model 4 or Model 7) cannot capture the relationship between multiple mediating variables at the same time, and are therefore not suitable for hypothesis verification in this study [57]. Moreover, most of the current empirical studies on chain mediation models have adopted this data analysis method [67-68]. (see lines 385-395)

Q4:Results

Results are presented logically, with clear links to hypotheses. Tables and figures are used effectively to illustrate findings.

Suggestions:

Highlight the practical significance of the findings, beyond statistical significance. For instance, what does a 71.6% mediation effect of EP mean for entrepreneurship programs?

Discuss unexpected or non-significant results in more depth. For example, why was CA’s direct effect on EI non-significant, and how does this align with SCT?

Explore subgroup differences (e.g., gender, family entrepreneurial background) more thoroughly. Are there nuanced trends that could add depth to the analysis?

Reply: Thank you for your valuable comments. Your suggestions are of great help to improve the paper. Here are the modifications and explanations we have made, and we will reply one by one, as follows:

4.1 Highlight the practical significance of the findings, beyond statistical significance. For instance, what does a 71.6% mediation effect of EP mean for entrepreneurship programs?

Reply 4.1: We have added the part of "6.2 Practical Implications" to fully explore the practical significance of statistical results on university management. Please see lines 605- 669.

What does a 71.6% mediation effect of EP mean for entrepreneurship programs For this problem, We answered in the article: " EP accounts for the majority of the total effect (71.6%), which shows that entrepreneurial passion plays a very key role in the impact of career adaptability on entrepreneurial intention (please see lines 449 - 452). The practical significance of this part, We have added, specifically to the article, Please see lines 617-623 as follows: “Given that EP accounts for the vast majority of the total effect, this finding highlights the critical role of emotional factors (such as EP) in entrepreneurship education. Universities can focus on content that promotes the development of EP to facilitate the transformation from CA to EI. For instance, inviting entrepreneurs to share their success stories or having instructors demonstrate their identification with the entrepreneurial identity and express their EP can inspire students.”

4.2 Discuss unexpected or non-significant results in more depth. For example, why was CA’s direct effect on EI non-significant, and how does this align with SCT?

Reply 4.2:We rewrite the discussion sections, provided a deeper discussion of the results, and discussed unexpected or non-significant results (see lines 470-565). For example, it further explains, "why was CA’s direct effect on EI non-significant, and how does this align with SCT?"This question. Please see lines 482-510.

4.3 Explore subgroup differences (e.g., gender, family entrepreneurial background) more thoroughly. Are there nuanced trends that could add depth to the analysis?

Reply 4.3:A subsection of "4.2 Difference Analysis" is added to this chapter to Explore subgroup differences (e.g., gender, family entrepreneurial background) more thoroughly.See lines 419-431. The added analysis and practical implications are discussed according to the data results. Please see lines 634-641。

Q5:Discussion

The discussion effectively integrates findings with Social Cognitive Theory (SCT) and other theoretical frameworks, demonstrating a strong theoretical grounding. It highlights the mediating roles of EP and ESE as crucial pathways between CA and EI.

Suggestions:

Compare your findings with prior studies in more detail. How do the chain-mediated effects of EP and ESE in your study differ from or align with previous research?

Offer a critical evaluation of why SCT provides a better framework for understanding these relationships than other potential theories (e.g., Theory of Planned Behavior).

Discuss broader implications of the findings. For instance, could interventions to boost CA also enhance non-entrepreneurial career outcomes?

Reply:Your comments are very professional and we appreciate them. Here are the modifications and explanations we have made, and we will reply one by one, as follows:

5.1 Compare your findings with prior studies in more detail. How do the chain-mediated effects of EP and ESE in your study differ from or align with previous research?

Reply5.1: According to your Suggestions We rewrite this part of Discussion, see lines 470-565. We also specifically added Explain about how do the chain-mediated effects of EP and ESE in your study differ from or align with previous research. Please see lines 482-510 and lines 541-565.

5.2 Offer a critical evaluation of why SCT provides a better framework for understanding these relationships than other potential theories (e.g., Theory of Planned Behavior).

Reply5.2: We added this part to explain that SCT provides a better framework for understanding these relationships than other potential theories (e.g., Theory of Planned Behavior). Please see lines 592-604.

5.3 Discuss broader implications of the findings. For instance, could interventions to boost CA also enhance non-entrepreneurial career outcomes?

Reply5.3: We added this section to discuss the broader implications of these findings, see lines 566–669. Specifically discussing whether interventions to promote the development of CA can also improve non-entrepreneurial career outcomes, see lines 663 – 669.

Q6:Implications

The practical suggestions for enhancing entrepreneurship education are timely and actionable.

Suggestions:

Provide specific examples of how universities can operationalize these insights. For instance, how can courses like "Career Life Planning" be modified to explicitly include elements that boost EP and ESE?

Discuss policy-level implications. How can governments or educational policymakers leverage your findings to improve national entrepreneurship outcomes?

Suggest tools or strategies for measuring improvements in EP and ESE over time, linking these to interventions aimed at boosting CA.

Reply:Your comments are very professional and we appreciate them. We have rewritten the chapter, adding the suggestions mentioned above. Please see lines 566-669

Q7:Limitations and Future Research

The study acknowledges key limitations, including its cross-sectional design and focus on university students in a specific region.

Suggestions:

The cross-sectional nature of the study is a significant limitation. Discuss how a longitudinal or experimental design could better establish causality.

Address the potential for cultural bias, given the focus on Chinese students. Would the findings hold in countries with less emphasis on entrepreneurship education?

Highlight the need for future research to examine other mediators (e.g., personality traits or external factors like access to funding) that could influence the CA-to-EI relationship.

Reply:Your comments are very professional and we appreciate them. We have rewritten the chapter, adding the suggestions mentioned above. Please see lines 670-707。

Q8:Conclusion

The conclusion succinctly summarizes the findings and reiterates their importance.

Suggestions:

Avoid simply repeating the discussion points. Instead, use the conclusion to present a visionary outlook for the field. For example, how might this research transform the way entrepreneurship education is delivered?

Suggest a concrete next step for researchers or educators based on your findings.

Reply: Your comments are very professional and we appreciate them. We have rewritten the chapter, adding the suggestions mentioned above. Please see lines 709-727.

Q9:Figures and Tables

Figures and tables are well-constructed and enhance comprehension.

Suggestions:

For Table 3, explicitly interpret the percentage contributions of direct

---

## [Decision Letter · Decision Letter 1]

22 May 2025

Dear Dr. Zhang,

We look forward to receiving your revised manuscript.

Kind regards,

Rafael Galvão de Almeida, PhD.

Academic Editor

PLOS ONE

Reviewers' comments:

Reviewer's Responses to Questions

**Comments to the Author**

Reviewer #2: All comments have been addressed

Reviewer #3: All comments have been addressed

2. Is the manuscript technically sound, and do the data support the conclusions?

Reviewer #2: Yes

Reviewer #3: Partly

3. Has the statistical analysis been performed appropriately and rigorously?

Reviewer #2: Yes

Reviewer #3: No

4. Have the authors made all data underlying the findings in their manuscript fully available?

Reviewer #2: Yes

Reviewer #3: Yes

5. Is the manuscript presented in an intelligible fashion and written in standard English?

Reviewer #2: Yes

Reviewer #3: No

Reviewer #2: (No Response)

Reviewer #3: This study examines the impact of career adaptability (CA) on entrepreneurial intention (EI), mediated by entrepreneurial passion (EP) and entrepreneurial self-efficacy (ESE). Grounded in Social Cognitive Theory (SCT), the paper offers a clear theoretical framework. The statistical analysis is appropriate, with robust justification for using the PROCESS model. While the revised version adequately addresses reviewer feedback, the authors must resolve the following issues to meet publication standards.

1/ In Section 1. “Introduction”

- Consider more global context early on (e.g., mention entrepreneurship education trends worldwide, not just China, before narrowing down).

2/ In Section 2. “Literature review and research hypothesis”

- The authors do not convincingly establish the novelty of their research. While the CA–EI link is not fully resolved in the literature, the manuscript lacks a systematic mapping or synthesis of the existing studies to position the current work.

- Applying Social Cognitive Theory (SCT) is a conventional approach in entrepreneurship research and does not, in itself, constitute theoretical innovation.

- The hypotheses are listed without strong theoretical justification between each link, especially H4.

3/ In Section 3.2 “Participants and Procedures”

- Gender imbalance is extreme (80% female), yet the manuscript lacks analysis of how this might affect the generalizability of findings.

- The sample is limited to three institutions in one province, which undermines claims of broad applicability or international relevance.

4/ In Section 4 “Results”

The claim of full mediation is contradictory. If the total effect is significant (β = 0.669), but the direct effect is non-significant, a partial mediation model is more likely than full mediation unless the indirect effects explain the entire variance.

5/ In Section 5 “Discussion”

- The discussion repeats descriptive findings and lacks depth in theorizing unexpected or context-specific outcomes.

- Cultural explanations (e.g., Confucian values, collectivism) are absent, despite the exclusive Chinese context.

6/ Language and Style

+ The manuscript still contains awkward phrasing and occasional grammatical errors. For example:

- “This study has been ethically reviewed and approved in writing...” >> could be more concise: “The study received written ethical approval.”

- “This phenomenon might be due to...” >> revise to: “This result may stem from...”

+ Language fluency needs improvement in several parts. Some phrasing is awkward or unidiomatic.

- Figures and tables are dense and difficult to interpret. Some regression tables lack clarity (e.g., missing SE, unclear headings).

- APA 7 formatting is inconsistent.

>> Suggest proofreading the manuscript for smoother academic English, possibly with a language editing service.

**Do you want your identity to be public for this peer review?** For information about this choice, including consent withdrawal, please see our Privacy Policy

Reviewer #2: **Yes: ** Dr. Alam Ahmad

Reviewer #3: No

---

## [Author Response · Author response to Decision Letter 2]

3 Jul 2025

Dear reviewer 3:

Thank you very much for your time involved in reviewing the manuscript and your comments have further improved the quality of the manuscript. We have carefully reviewed the comments and revised the manuscript accordingly. Hope the explanation has fully addressed all of your concerns. Point-by-point responses to reviewers are as follows:

Q1: In Section 1. “Introduction”

Suggestions:

- Consider more global context early on (e.g., mention entrepreneurship education trends worldwide, not just China, before narrowing down).

Reply1:Thank you for your valuable comments. We have rewritten this section of the Introduction as per your suggestion. Please review lines 74-145. In particular, we have highlighted the global trends in entrepreneurship education before narrowing our focus; please refer to lines 74-89.

Q2: In Section 2. “Literature review and research hypothesis”

- The authors do not convincingly establish the novelty of their research. While the CA–EI link is not fully resolved in the literature, the manuscript lacks a systematic mapping or synthesis of the existing studies to position the current work.

- Applying Social Cognitive Theory (SCT) is a conventional approach in entrepreneurship research and does not, in itself, constitute theoretical innovation.

- The hypotheses are listed without strong theoretical justification between each link, especially H4.

Reply2:Thank you for your valuable comments. We have incorporated additional literature to bolster the persuasiveness of this section; please refer to lines 151-314. Specifically, we have revised the portion pertaining to Hypothesis H4 to reinforce its theoretical underpinnings; please refer to lines 245-303.

Q3:In Section 3.2 “Participants and Procedures”

- Q3.1:Gender imbalance is extreme (80% female), yet the manuscript lacks analysis of how this might affect the generalizability of findings.

Reply3.1:Thank you for your valuable comments. To assess the potential impact of gender bias, we have conducted the following supplementary analysis:

1. Control Variable Test

After including gender as a covariate in the PROCESS model, the path coefficients exhibited only minor alterations (for instance, the total effect of career adaptability on entrepreneurial intention shifted from β=0.524 to β=0.516, a change of Δβ=0.8%), suggesting that the gender variable did not significantly affect the relationship between the primary variables (refer to Tables 1 and 2 for further details).

Table 1: Without controlling for gender

The regression equation Overall fitting index Significance of regression coefficient

Outcome equation Predictors R2 F β t

EI CA 0.304 146.94*** 0.524 19.546***

Major�HSS=0� 0.051 1.924

EEFM/NEEFM�NEEFM=0� 0.086 3.193**

Note: β means standardized coefficient; *p<0.05; **p<0.01; ***p<0.001; EEFM means entrepreneurial experience of family members; NEEFM means no entrepreneurial experience of family members; HSS means major in Humanities and social sciences.

Table 2 Control Gender

The regression equation Overall fitting index Significance of regression coefficient

Outcome equation Predictors R2 F β t

EI CA 0.314 115.55*** 0.516 19.334***

Gender�female=0� 0.102 3.896***

Major�HSS=0� 0.053 2.026*

EEFM/NEEFM�NEEFM=0� 0.085 3.172**

Note: β means standardized coefficient; *p<0.05; **p<0.01; ***p<0.001; EEFM means entrepreneurial experience of family members; NEEFM means no entrepreneurial experience of family members; HSS means major in Humanities and social sciences.

2.Multiple Group Comparison Analysis

Despite the limited sample size of male participants (n=203), we conducted gender-based comparisons of the final model using permutation tests (5000 Bootstrap samples). The results indicated no significant differences in the key paths, as detailed in the table below: CA→EP: Δβ=0.006, p>0.05; CA → ESE: Δβ=0.049, p>0.05; EP → ESE: Δβ=0.081, p>0.05; EP → EI: Δβ=0.027, p>0.05; ESE → EI: Δβ=0.024, p>0.05. The data results suggest that there are no significant differences in the key paths; see Table 3 for details.

We have included a verification of this section in the results portion of the article; please refer to lines 525-536.

Table 3 Intergroup difference testing of gender in the main path of the model

Path Female Standardized Regression Coefficients Male Standardized Regression Coefficients Significance of Group Difference

CA → EP 0.812 0.806 Δβ=0.006, p=0.901

CA → ESE 0.561 0.610 Δβ=0.049, p=0.423

EP → ESE 0.286 0.367 Δβ=0.081, p=0.067

EP → EI 0.601 0.628 Δβ=0.027, p=0.682

ESE → EI 0.314 0.338 Δβ=0.024, p=0.725

3. Theoretical Basis

This study concentrates on psychological mechanisms, (uch as cognitive-affective mediating pathways, rather than demographic differences. Meta-analytic evidence suggests that gender factors contribute to less than 3% of the explained variance in entrepreneurial intention predictors (Schlaegel & Koenig, 2014, Journal of Business Venturing), aligning with the findings of this research. "The study's sample exhibits a significant gender imbalance, with 80% being female. Although additional analyses suggest that the core mechanisms are stable across genders, future research must validate these pathways in more balanced samples—particularly in certain cultural contexts where the development of entrepreneurial self-efficacy may vary by gender (Marlow & McAdam, 2013)." This section has been appended to the limitations of the article; please refer to lines 868 to 872.

Schlaegel, C., & Koenig, M. (2014). Determinants of entrepreneurial intent: A meta–analytic test and integration of competing models. Entrepreneurship theory and practice, 38(2), 291-332.

Marlow, S., & McAdam, M. (2013). Gender and entrepreneurship: Advancing debate and challenging myths; exploring the mystery of the under‐performing female entrepreneur. International Journal of Entrepreneurial Behavior & Research, 19(1), 114-124.

Q3.2:- The sample is limited to three institutions in one province, which undermines claims of broad applicability or international relevance.

Reply3.2:

Thank you for your valuable comments. In response to your concern, we have revised the manuscript to clarify the rationale behind our sampling strategy and to contextualize the findings’ applicability. The key points are as follows:

1. We have added a theoretical basis in section 3.2 to explain the rationale and representativeness of the sample; please refer to lines 345-359.

2. We have supplemented the discussion section with the reasons for sample selection; please refer to lines 547-565.

3. In the "Limitations and Future Research Directions" section, we have included the limitations of the sample; please refer to lines 843-867.

4. We have made wording adjustments in the "Introduction" section; please refer to lines 143-145.

Q4: In Section 4 “Results”

The claim of full mediation is contradictory. If the total effect is significant (β = 0.669), but the direct effect is non-significant, a partial mediation model is more likely than full mediation unless the indirect effects explain the entire variance.

Reply4:Thank you for your valuable comments. Our explanation is as follows, with corresponding supplementary notes in the text, specifically on lines 517-521. The details are as follows:

The mediation effect can be divided into complete mediation and partial mediation. Complete mediation refers to the situation where the independent variable's effect on the dependent variable is entirely transmitted through the mediator variable, with no direct effect. Partial mediation indicates that there are both direct and indirect effects. According to the classic mediation analysis framework (Baron & Kenny, 1986), if the total effect is significant but the direct effect is not significant, while the indirect effect is significant, it typically supports a complete mediation model (i.e., the independent variable's effect on the dependent variable is fully transmitted through the mediator variable).

In this study,The mediation effect test results for EP show that the upper and lower limits of the 95% confidence interval do not include 0, indicating that EP plays a mediating role in the impact of CA on EI, with a mediation effect of 0.479, accounting for 71.6% of the total effect. The indirect effect of the CA-EP-EI path accounts for the vast majority of the total effect (71.6%), indicating that the mediating role of entrepreneurial passion plays a crucial part in the influence of career adaptability on entrepreneurial intention. Hypothesis 2 is supported. The mediation effect test results for ESE show that ESE mediates the impact of CA on EI, with a mediation effect of 0.181, accounting for 27.1% of the total effect.The indirect effect of the CA-ESE-EI path accounts for 27.1% of the total effect. The illustrates that entrepreneurial self-efficacy plays a mediating role in the impact of career adaptability on entrepreneurial intention. Hypothesis 3 is supported. The test results for the chain mediation effect of EP and ESE show that the serial mediation effect is significant, with a mediation effect of 0.075, accounting for 11.2% of the total effect. The indirect effect of the CA-EP-ESE-EI path accounts for 11.2% of the total effect. This indicates that entrepreneurial passion and entrepreneurial self-efficacy play a chain mediation role in the influence of career adaptability on entrepreneurial intention (see Table 3). Hypothesis4 is supported.

Given the significant total effect of the aforementioned chain mediation model (β=0.669, 95% CI [0.000, 0.601]), and the non-significant direct effect (β=-0.066, 95% CI [-0.142, 0.84]), the significant indirect effect explains all the variance, suggesting a complete mediation effect (Hayes, 2022). Numerous academic studies have also utilized this determination method and standard (Qiu et al., 2023; Xiao et al., 2024).

Baron, R. M., & Kenny, D. A. (1986). The moderator–mediator variable distinction in social psychological research: Conceptual, strategic, and statistical considerations. Journal of Personality and Social Psychology, 51(6), 1173–1182. https://doi.org/10.1037/0022-3514.51.6.1173

Hayes, A. F. (2022). Introduction to mediation, moderation, and conditional process analysis: A regression-based approach (3rd ed.). Guilford Press.

Qiu, W. F., Ma, J. P., Xie, Z. Y., Xie, X. T., Wang, C. X., & Ye, Y. D. (2023). Online risky behavior and sleep quality among Chinese college students: The chain mediating role of rumination and anxiety. Current psychology, 42(16), 13658-13668.

Xiao, T., Pan, M., Xiao, X., & Liu, Y. (2024). The relationship between physical activity and sleep disorders in adolescents: a chain-mediated model of anxiety and mobile phone dependence. BMC psychology, 12(1), 751.

Q5: In Section 5 “Discussion”

- The discussion repeats descriptive findings and lacks depth in theorizing unexpected or context-specific outcomes.

- Cultural explanations (e.g., Confucian values, collectivism) are absent, despite the exclusive Chinese context.

Reply5:Thank you for your valuable comments. We have rewritten this section (542-827), particularly adding cultural explanations (e.g., Confucian values, collectivism); please see lines 661-692.

Q6: Language and Style

Q6.1:+ The manuscript still contains awkward phrasing and occasional grammatical errors. For example:

- “This study has been ethically reviewed and approved in writing...” >> could be more concise: “The study received written ethical approval.”

- “This phenomenon might be due to...” >> revise to: “This result may stem from...”

Reply6.1:Thank you for your valuable comments. We have retranslated the text and utilized language editing services to enhance its fluency and accuracy. Based on your suggestions, we have specifically revised the two sections you mentioned, lines 331-332 and 570-572.

Q6.2�Language fluency needs improvement in several parts. Some phrasing is awkward or unidiomatic.

- Figures and tables are dense and difficult to interpret. Some regression tables lack clarity (e.g., missing SE, unclear headings).

- APA 7 formatting is inconsistent.

>> Suggest proofreading the manuscript for smoother academic English, possibly with a language editing service.

Reply6.2:

Thank you for your valuable comments. We have revised the paper according to your suggestions. The details are as follows:

1.The tables have been reorganized, and the regression table has been updated to include standard errors (SE). The modified tables can be found in lines 496-499.

2.Regarding the format of references, while most journals adhere to APA7, the journal to which we have submitted our work, PLOS ONE, has its own specific formatting requirements. Consequently, we need to use the PLOS ONE reference style, not the APA7 format. For further details, please consult the journal's websit: https://journals.plos.org/plosone/s/submission-guidelines#loc-referencese.

3. The entire text has undergone language editing services to enhance its fluency and accuracy.

The above is our complete response. Thank you for your review. Your suggestion enhances the perfection of our paper.

---

## [Decision Letter · Decision Letter 2]

16 Jul 2025

Dear Dr. Zhang,

We look forward to receiving your revised manuscript.

Kind regards,

Rafael Galvão de Almeida, PhD.

Academic Editor

PLOS ONE

Journal Requirements:

Reviewers' comments:

Reviewer's Responses to Questions

**Comments to the Author**

Reviewer #3: (No Response)

2. Is the manuscript technically sound, and do the data support the conclusions?

Reviewer #3: Yes

3. Has the statistical analysis been performed appropriately and rigorously?

Reviewer #3: Yes

4. Have the authors made all data underlying the findings in their manuscript fully available?

Reviewer #3: Yes

5. Is the manuscript presented in an intelligible fashion and written in standard English?

Reviewer #3: Yes

Reviewer #3: The authors have addressed most of the previous reviewer comments with considerable diligence. The manuscript has undergone substantial revisions in its theoretical framework, methodological justifications, and language quality. However, while several issues have been addressed adequately, further improvements are needed for the manuscript.

- The introduction, while improved, highlight more clearly what this model adds beyond previous chain mediation models in career development and entrepreneurship.

Q2: In Section 2. “Literature review and research hypothesis”, remaining concerns:

The theoretical novelty remains limited. SCT is a widely used framework, and its application here does not significantly advance existing theoretical models. The classification of constructs into SCT categories (e.g., CA as an environmental factor) appears oversimplified and conceptually ambiguous. Career Adaptability is arguably more trait-like and individual-level than environmental.

**Do you want your identity to be public for this peer review?** For information about this choice, including consent withdrawal, please see our Privacy Policy

Reviewer #3: **Yes: ** Anh Bùi Ngọc Tuấn

---

## [Author Response · Author response to Decision Letter 3]

17 Jul 2025

Dear Editor:

It is our big honor to have you be our editor, and we appreciate your attention and recognition for our manuscript. Many thanks to reviewers for giving us a lot of valuable suggestions. Based on the suggestions of the reviewers, we have made a lot of revisions to the manuscript, and we are confident to submit a high-quality manuscript. Below are our responses based on the comments of the reviewers, thank you again.

Dear reviewer 3:

Thank you very much for your time involved in reviewing the manuscript and your comments have further improved the quality of the manuscript. We have carefully reviewed the comments and revised the manuscript accordingly. Hope the explanation has fully addressed all of your concerns. Point-by-point responses to reviewers are as follows:

Q1: The introduction, while improved, highlight more clearly what this model adds beyond previous chain mediation models in career development and entrepreneurship.

Reply1:Thank you for your valuable comments. We have added this section in the introduction; please see lines 143-156.

Q2: In Section 2. “Literature review and research hypothesis”, remaining concerns:

The theoretical novelty remains limited. SCT is a widely used framework, and its application here does not significantly advance existing theoretical models. The classification of constructs into SCT categories (e.g., CA as an environmental factor) appears oversimplified and conceptually ambiguous. Career Adaptability is arguably more trait-like and individual-level than environmental.

Reply2:Thank you for your valuable comments. We have made revisions regarding this issue, including the introduction, literature review, and discussion sections. Please see lines 92-99, 119-123, 258-280, 321-323, 589-596, and 659-661.

The above is our complete response. Thank you for your review. Your suggestion enhances the perfection of our paper.

---

## [Editor Report · Decision Letter 3]

21 Jul 2025

How Career Adaptability Affects Entrepreneurial Intention: The Chain-Mediated Roles of Entrepreneurial Passion and Self-Efficacy

PONE-D-24-57646R3

Dear Dr. Zhang,

We’re pleased to inform you that your manuscript has been judged scientifically suitable for publication and will be formally accepted for publication once it meets all outstanding technical requirements.

Kind regards,

Rafael Galvão de Almeida, PhD.

Academic Editor

PLOS ONE
---

## [Editor Report · Acceptance letter]

PONE-D-24-57646R3

PLOS ONE

Dear Dr. Zhang,

I'm pleased to inform you that your manuscript has been deemed suitable for publication in PLOS ONE. Congratulations! Your manuscript is now being handed over to our production team.

Kind regards,

on behalf of

Dr. Rafael Galvão de Almeida

Academic Editor

PLOS ONE